# A continuum of bright and dark-pulse states in a photonic-crystal resonator

Su-Peng Yu [1,2✉], Erwan Lucas[1,2], Jizhao Zang[1,2] & Scott B. Papp[1,2✉]

Nonlinearity is a powerful determinant of physical systems. Controlling nonlinearity leads to interesting states of matter and new applications. In optics, diverse families of continuous and discrete states arise from balance of nonlinearity and group-velocity dispersion (GVD). Moreover, the dichotomy of states with locally enhanced or diminished field intensity depends critically on the relative sign of nonlinearity and either anomalous or normal GVD. Here, we introduce a resonator with unconditionally normal GVD and a single defect mode that supports both dark, reduced-intensity states and bright, enhanced-intensity states. We access and explore this dark-to-bright pulse continuum by phase-matching with a photonic-crystal resonator, which mediates the competition of nonlinearity and normal GVD. These stationary temporal states are coherent frequency combs, featuring highly designable spectra and ultralow noise repetition-frequency and intensity characteristics. The dark-to-bright continuum illuminates physical roles of Kerr nonlinearity, GVD, and laser propagation in a gapped nanophotonic medium.

[1] Time and Frequency Division, National Institute of Standards and Technology, Boulder, CO, USA. [2] Department of Physics, University of Colorado, Boulder, CO 80309, USA. ✉email: supeng.yu@colorado.edu; scott.papp@nist.gov

Complex systems generate patterns from a fundamental set of rules, which govern interactions between system components. Fractals are a good example[1] in which mathematical relations produce intricate patterns, depending on a set of parameters that characterize the relations. Similarly, in nonlinear optics, spatiotemporal laser patterns readily manifest in a medium, and their dynamics enable detailed, precise, and controllable tests of how light and matter interact. We focus on a ubiquitous nonlinearity of materials, the Kerr effect[2,3], which underlies fascinating behaviors in nonlinear optics, such as the formation of discrete states of patterns and pulses[4,5]. However, the Kerr effect only represents half of the picture in the dynamics, since the frequency dependence of group-velocity dispersion (GVD or simply dispersion) typically controls what optical state forms at the balance against nonlinearity. Simply the sign of GVD differentiates optical states, for example in the case of anomalous GVD that balances with the Kerr effect for soliton formation. Beyond illuminating complex systems, nonlinear-optical states are being applied and optimized. Dissipative Kerr solitons in microresonators enable ultraprecise optical-frequency metrology[6] and many other functionalities[7–11]. More advanced device topologies promise to yield enhanced nonlinear laser sources, through coupled resonators[12,13], dispersion engineering in nanophotoncis[14,15], and inverse-design methods[16,17].

To understand the interplay of nonlinearity and either anomalous or normal GVD, we utilize the mean-field Lugiato-Lefever equation (LLE)[18] for the field in a Kerr resonator. The LLE describes several states: the flat state of a sufficiently low-intensity pump laser; oscillatory Turing patterns that extend over the entire resonator; and localized bright and dark solitons, which are the canonical stationary states at anomalous and normal GVD, respectively. In particular, the LLE framework for optical states subject to normal GVD (and a positive nonlinear coefficient) is relatively sparse, since the nearly unconditional imbalance of GVD and nonlinearity at constant excitation suppresses phase matching[19]. Still, particular nonlinear states have been observed through fortuitous mode-structure defects that create bands of anomalous GVD in an otherwise normal-GVD resonator to seed dark-soliton formation[10,12]. Details on normal GVD phase matching and the comb phases thus created are presented in Supplementary Section I and explored in[20].

Experiments with dark-soliton states exhibit strikingly different behavior from its anomalous GVD counterpart, for example, the development of complex spectral modifications with pump-laser detuning[13]. Moreover, effectively bright-pulse states such as the platicon have been described through simulation[20] and can form through multi-frequency pumping[21] or the Raman effect[22]. Beside interesting physics, normal GVD systems are advantageous for applications, including the relative ease of obtaining normal GVD, self-starting pulses[20], focused spectral power distribution, and high energy efficiency[23]. These characteristics provide complementary functionalities to the current paradigm for laser synthesis[24] and optical clockwork[6] with anomalous dispersion Kerr combs. It is therefore important to understand the physics underlying the emergence of normal GVD solitons, and to develop reliable methods to create these curious states.

Here, we phase-match for pattern formation in normal GVD, using a photonic-crystal resonator (PhCR)[25]. The tailored frequency-domain point-defect in PhCR dispersion enables us to discover and assess a complete continuum of bright- and dark-pulse states under normal GVD. The PhCR defect modifies the pump-versus-loss energy balance within the resonator, providing the key that uncovers dark- and bright-pulse states in the same physical device. A PhCR is a microresonator with periodic modulation that demonstrates Bloch symmetry[26], opening a bandgap in the resonator dispersion. We use an edge-less boundary condition– an azimuthally uniform pattern around the resonator –to create the frequency-domain equivalent of a point-defect on a targeted azimuthal mode of the resonator[27]. By tuning a pump laser onto resonance with this point-defect mode, we phase-match for modulation instability in the normal GVD regime. Further, by designing the PhCR bandgap for specific regimes of pump laser power and detuning, we can realize both bright- and dark-pulse states of the resonator field. Our experiments explore their tuning behavior with the pump laser and bandgap, establishing a full continuum between the bright-pulse or 'platicon' states[21,22] and dark-pulse states[10,12]. Moreover, we characterize the utility of states in the dark-to-bright soliton continuum for applications through ultraprecise optical-frequency measurements.

## Results

**Phenomenon.** Unifying and controlling the available states in normal-GVD Kerr resonators is an important objective; see Fig. 1. The bandgap of our PhCR devices enables a controllable frequency shift of the mode excited by the pump laser, which unconditionally satisfies phase matching for four-wave mixing. In experiments on this system, we observe spontaneous formation of optical states with spectra that suggest localized patterns. Analyzing these states highlights the curious characteristic in which the high-intensity duration $\tau$ (Fig. 1a) varies dramatically with the laser detuning $\alpha = \omega_r - \omega_l$, where $\alpha > 0$ for laser frequency $\omega_l$ lower than the resonance $\omega_r$. Indeed, we identify these states as pulses that transition continuously from dark to bright for a prescribed tuning range of $\alpha$. We characterize this by the intensity-filling fraction $t_c = \tau/\tau_{rep}$, which is the pulse duration normalized to the round-trip time $\tau_{rep}$ or equivalently the fraction of the azimuthal angle $\theta$ occupied by higher-than-average optical intensity. Figure 1b illustrates this dark-to-bright continuum with an accurate LLE simulation for a PhCR device. The PhCR shift modifies the phase-matching condition and therefore what states can be reached in the resonator[25]. Here we choose a pump-laser mode (number $\mu = 0$) red-shifted by 2.0 $\kappa$ from the baseline normal GVD, where the half-linewidth of the resonator is $\kappa/2\pi$. We simulate an $\alpha$ sweep with fixed pump power and mode shift. The plot shading indicates the continuous evolution between diminished and enhanced peak intensity, state i and iii in Fig. 1b, respectively. The intermediate state ii represents a half-filled resonator.

Figure 1c, i–iii analyze specific states in the dark-to-bright soliton continuum, making the connection between simulated spectra and pulse waveforms and our spectrum measurements of states created in normal-GVD PhCR devices. As a function of $\alpha$ starting near zero, the stationary state begins as a dark soliton[12] where a localized intensity dip exists over an otherwise flat background, i.e. the anti-pulse state i. The intensity dip creates an effective negative Kerr constant in normal GVD, and approximates inverted Kerr soliton temporal shape and near $sech^2$ spectral envelope. As $\alpha$ increases, the anti-pulse grows in duration and develops multiple intensity minima that extend about $\theta$[13]. State ii shows the dark pulse with five minimums and occupying approximately half of the $\theta$ space within the resonator. We equivalently interpret this half-filled state as a bright pulse occupying the other half of the resonator. Increasing $\alpha$ further causes the bright section to shorten temporally and increase in intensity to form the platicon state[20,21], which phenomenologically describes this bright-pulse state. Indeed platicons are described by their plateau-like temporal shape and localized oscillations trailing the pulse (state iii). Our results provide the link between the bright- and dark-pulse states as the two extremes of a continuously tuned intensity pattern.

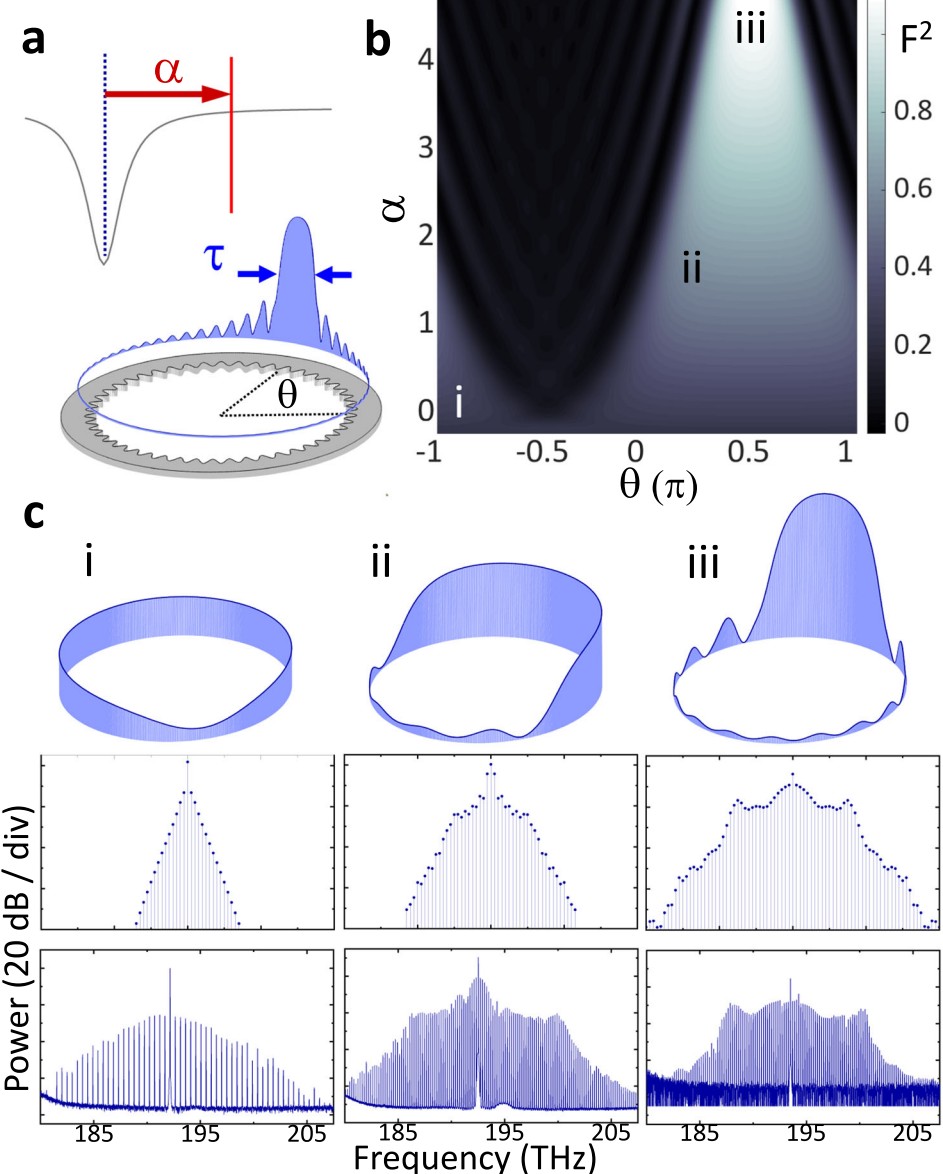

**Fig. 1 Phenomena of the pulse continuum. a** Illustration of the relation between detuning $\alpha$ and pulse width $\tau$. **b** Continuous pump detuning sweep showing **i** dark pulse, **ii** half-filled, and **iii** bright-pulse states. **c** Simulated intensities (top), spectra (middle), and measured spectra (bottom) of each state **i–iii**.

According to the intrinsic temporal- and spectral-domain relationships of nonlinear states, the pulse waveforms spanning the dark-to-bright pulse continuum in Fig. 1 exhibit identifying features. We focus on three primary features according to their spectral-domain appearance: center lobe, wing, and horn. The center lobe refers to the high-power modes near the pump laser, and its bandwidth inversely represents the temporal size of a bright or dark localized pulse. Indeed, both dark- (Fig. 1i) and bright-pulse states (Fig. 1iii) prominently feature a center lobe, and the prototype dark soliton is almost entirely composed of it. The more exotic wing feature develops outside the bandwidth of the center-lobe at larger $\alpha$. It represents the deviation of the temporal pattern from a pulse and is most visible when the pattern is temporally extended (Fig. 1ii), reminiscent of the spectrum of a bandwidth-limited square wave. Indeed, the power spectrum of the half-filled state is primarily composed of the $\mu = \pm 1$ modes while the wings feature a $1/\mu^2$ asymptotic decay, in agreement with that of a periodic rectangular waveform. More-over, the half-filled state demarcates the condition of center-lobe broadening and temporal pulse compression either to a dark or

bright state with any change to $\alpha$. This behavior stands apart from anomalous dispersion Kerr solitons[19,28] in which increasing $\alpha$ leads to a monotonic increase in the soliton bandwidth while approximately maintaining the power-per-line near the pump[29]. The horn feature refers to the heightened spectral power on the edges of the spectral bandwidth (Fig. 1iii). It represents the rapid oscillation trailing the pulse, a normal GVD correspondence of the dispersive waves[30]. More information about features of the normal GVD soliton is provided in Supplementary Fig. S1. These features help us extract waveform information from the optical spectra in the following sections.

**Mechanism**. We develop a theoretical framework for the pulse-duration-tuning behavior throughout the dark- to bright-pulse continuum. Specifically, we establish the relation between the intensity-filling ratio $t_c$ and the laser detuning $\alpha$. Normal-GVD waveforms consist of two intensity levels, which depend on $\alpha$ and the pump $F$. The two levels are connected by switching fronts[31]. Here, we show that the PhCR disturbs these levels through an

effective pump contribution $F'$, which introduces an additional energy exchange between the two levels that determines $t_c$.

We begin exploring the relation $t_c(\alpha)$ by characterizing the waveform under normal GVD, using the pump-shifted LLE (hereafter PS-LLE)[25] that accurately describes the field of the PhCR:

$$\partial_t \psi = -(1 + i\alpha)\psi - \frac{i\beta}{2}\partial_\theta^2 \psi + i|\psi|^2 \psi + F + i\epsilon(\bar{\psi} - \psi), \quad (1)$$

where $\bar{\psi}$ denotes the average field over $\theta$, and $\beta = -2D_2/\kappa$ is the normalized dispersion parameter. For time-stationary solutions, $\partial_t \psi = 0$, the interaction between dispersion and nonlinearity manifests in the imaginary part of this equation. This yields in absence of the PhC term (assuming a real-values pulse $\psi$ for simplicity): $\frac{-\beta}{2}\partial_\theta^2 \psi/\psi + |\psi|^2 \simeq \alpha$. For example, in anomalous GVD ($\beta < 0$), the peak of a pulse shows negative curvature and positive Kerr shift. Partial cancellation or balance between the two enables sharp waveforms like the Kerr soliton. Our system is in normal GVD ($\beta > 0$), leading to competition between dispersion and nonlinearity, which sum to a particular $\alpha$ to phase-match to the pump laser. This leads to a mutually exclusive relation between local intensity and curvature—where the intensity is high, the waveform is flat—leading to the flat-top waveform with switching edges in between shown in Fig. 2a. $t_c$ corresponds to the fraction of the high-intensity level centered at $\theta = 0$ and $1 - t_c$ to the fraction of the low-intensity level in the remaining space in the resonator.

This two-level waveform already exists in the normal-GVD regime of conventional resonators[32]. The waveform corresponds to the flat-amplitude levels of the bi-stable, continuous-wave (CW) resonator field[31], controlled by $F$ and $\alpha$ through solutions to $F^2 = (1 + (|h|^2 - \alpha)^2)|h|^2$[19], where formally $h = F/(1 + i(\alpha - |h|^2))$ is the field at each level. These levels are stationary in time because the loss $\rho_{loss}$ and pump power in-flow $\rho_{in}$ per unit $\theta$ balance

$$\rho_{loss}(\theta) = \kappa \cdot I(\theta) = |\psi(\theta)|^2 \quad (2)$$

$$\rho_{in}(\theta) = F \cdot \mathbb{R}e(\psi(\theta)) \quad (3)$$

where $I(\theta)$ is the intensity, $\kappa = 1$ is the normalized loss rate; see Supplementary Section II. In a conventional, normal-GVD resonator, the balance $\rho_{in} = \rho_{loss}$ is satisfied locally for all $\theta$ at the bi-stability levels. Therefore, switching edges can translate independently about $\theta$ without perturbing the input-output energy flow, although weak oscillating tails of the edges can trap some waveforms in a specific configuration[32].

The mode structure of a PhCR perturbs the energy balance of the two levels, leading to curious nonlinear dynamics. We identify the impact of the PhCR frequency shift term in Eq. (1) by casting it in a conventional LLE with effective pump parameters $F'$, $\alpha'$:

$$F' = |F + i\epsilon\bar{\psi}| \quad (4)$$

$$\alpha' = \alpha + \epsilon \quad (5)$$

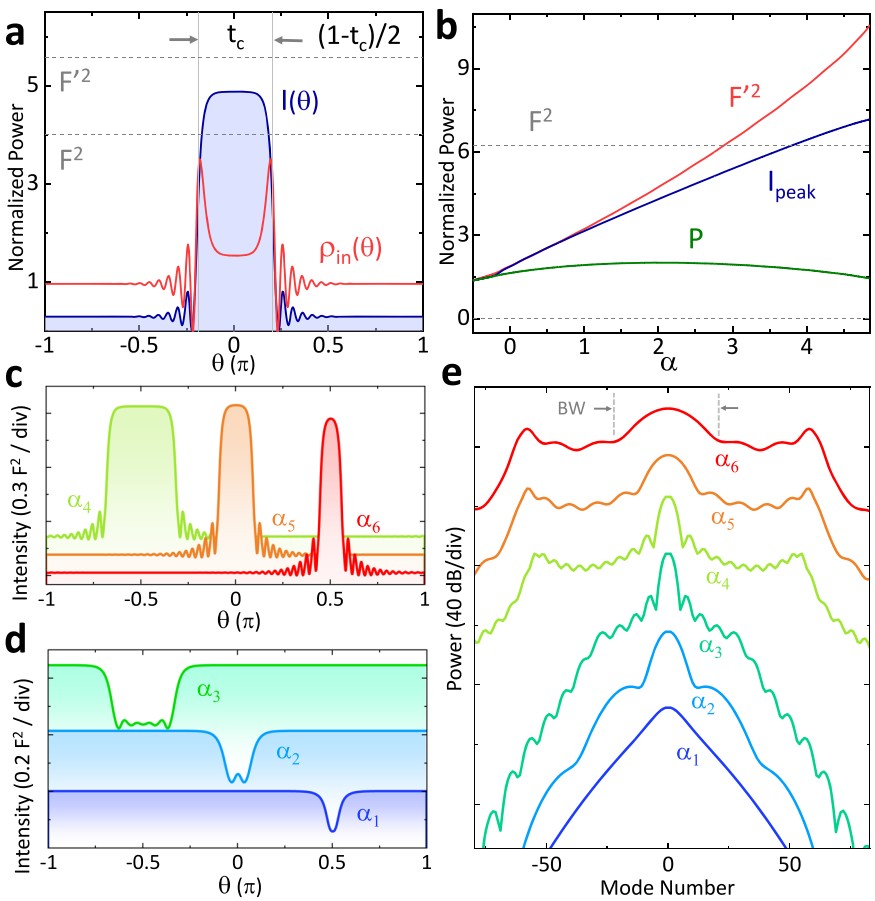

**Fig. 2 Theory and simulation. a** Intensity profile $I(\theta)$ of a waveform with bright-pulse fraction $t_c$. $F^2 < I(\theta) < F'^2$ for the high-intensity level, where $F$ and $F'$ are the pump and effective pump parameters. The pump energy in-flow $\rho_{in}(\theta)$ is also shown. **b** Calculated parameters $F'^2$, peak intensity $I_{peak}$, and pump power $P$ in the LLE. Note that $I_{peak} > F^2$ at high $\alpha$. **c, d** Plot the waveform on the red- and blue-detuned ranges of an $\alpha$ sweep ($\alpha_1 < \alpha_2 \cdots < \alpha_6$), showing the increase of dark-pulse duration and the decrease of bright-pulse duration with $\alpha$. The corresponding spectra are shown in (**e**). Note the behavior of the center lobe bandwidth BW marked in gray.

where the corresponding field amplitudes within each intensity level (neglecting the effect of curvature) are written as

$$h = \frac{F'}{1 + i(\alpha' - |h|^2)}, \qquad (6)$$

where $|h|^2$ approximates the high- and low-level intensities at $\theta = 0$ and $\pi$ in Fig. 2a. With $h$ determined by $F'$ but $\rho_{in}$ still dependent on the physical pump $F$, $\rho_{in}$ and $\rho_{loss}$ are no longer equal (these rates have the same sign by our definition). In the PhCR case, $h$ at the high-intensity level is driven to be larger by the effective $F'$. This is the mechanism that creates a larger peak intensity, forming a bright pulse.

Figure 2a shows $\rho_{in}(\theta)$ and $\rho_{loss}(\theta) = \kappa I(\theta)$, calculated using the PS-LLE. The high-intensity level exhibits a deficit of energy, $I > \rho_{in}$, while the low-intensity level shows an energy surplus, $I < \rho_{in}$. When combined, the two levels maintain energy conservation. The surplus or deficit arises from the difference in the relations $I(\theta) \propto |h|^2$ and $\rho_{in} \propto |h|^1$, which increase with $\alpha$ and $F'$. Importantly, both $F' > F$ and high-level intensity $I(0) = |h|^2 > F^2$ can be reached at large detuning. Figure 2b shows the behavior of $F'$ and $I_{peak}$ for the same parameters as Fig. 1b, where the peak intensity $I_{peak}$ in the PS-LLE corresponds to the high-level $I(0)$. We see both parameters surpass $F^2$ at large $\alpha$. To maintain the $I_{peak} > F^2$ intensity with the limited physical pump power $F^2$ available to the system, the pulse duration reduces as its intensity increases. This manifests as reducing $t_c$ (hence reducing the energy deficit) and also regularizing the increase of $F'$ through the $\bar{\psi} \simeq h \cdot t_c$ dependence. As result, the total energy inflow $P = \oint \rho_{in}(\theta) d\theta$ in Fig. 2b remains relatively constant in contrast to the increase of $F'$ or $I_{peak}$. This energy balance links $t_c$ to the peak field $h$, and thus to $\alpha$.

We calculate $t_c$ through the energy balance by integrating all energy flow within the resonator. We obtain the energy conservation condition by multiplying Eq. (1) by $\psi^*$, followed by integrating the terms over $\theta$:

$$(1 + i\alpha) \oint |\psi|^2 d\theta = \frac{i\beta}{2} \oint |\partial_\theta \psi|^2 d\theta + i \oint |\psi|^4 d\theta \\ + F\bar{\psi}^* + i\epsilon (|\bar{\psi}|^2 - \oint |\psi|^2 d\theta) \qquad (7)$$

where the second-derivative in $\theta$ term is integrated by part. Taking the real part of this form, we obtain the energy-balance equation:

$$\oint |\psi|^2 d\theta = \oint F \cdot \mathbb{R}e(\psi) d\theta \qquad (8)$$

where we identify the terms on the two sides corresponding to $\oint I(\theta) d\theta = \oint \rho_{in}(\theta) d\theta$. Expressing this form approximately in terms of the fields at $\theta = 0, \pi$ and $t_c$, we obtain

$$t_c \cdot I(0) + (1 - t_c) \cdot I(\pi) = t_c \cdot \rho_{in}(0) + (1 - t_c) \cdot \rho_{in}(\pi), \qquad (9)$$

which we rearrange to $t_c = \frac{\rho_{in}(\pi) - I(\pi)}{\rho_{in}(\pi) - \rho_{in}(0) + I(0) - I(\pi)}$. This form indicates how $t_c$ depends explicitly on the intensities of the two levels, and therefore implicitly on $\alpha$.

Figure 2c, d presents time-domain PS-LLE solutions across the continuum at $\beta = 5.2 \times 10^{-3}$, describing the soliton as we vary $\alpha$ to access both dark and bright pulses. We tune $\alpha$ from a setting $\alpha_1$ that yields the dark pulse, to the longer-duration dark pulse $\alpha_3$, across the half-filled state to a setting that yields the bright pulse $\alpha_4$, then to shortening bright pulse $\alpha_6$. The PS-LLE calculations confirm the monotonic tuning of $t_c$ with $\alpha$ described in the mechanism. Fig. 2e shows the corresponding spectra for $\alpha_1$ through $\alpha_6$. We highlight how the center-lobe bandwidth BW is governed by $t_c$. The monotonic increase of $t_c$ manifests as a lengthening dark pulse ($\alpha_1$ through $\alpha_3$) and reducing BW, but as a

shortening bright pulse ($\alpha_4$ through $\alpha_6$) and increasing BW with $\alpha$. See Supplementary Section III for details. The tuning of the center-lobe bandwidth will be the parameter we measure in our experiments.

**Experiment**. We fabricate normal GVD PhCR devices to explore the dark-to-bright pulse continuum. Our objectives are to: create devices with the GVD and $\epsilon$ settings that coincide with our theoretical predictions; energize the devices with a range of $\alpha$ settings to create optical states; and identify the spectral characteristics of these states. PhCRs are ring resonators with a sinusoidal modulation of the inner edge. The modulation amplitude determines the frequency shift $\epsilon$ of one mode. We couple the PhCR devices evanescently with a waveguide on the chip, illustrated in Fig. 3a. We anticipate that normal-GVD solitons arise spontaneously from instability of the flat state[20], and the out-coupled laser pulse forms a frequency comb, which we characterize through spectral-domain measurements. Time- and spectral-domain measurements show good correspondence for normal dispersion patterns[12]. We infer time-domain characteristics from the spectra and their tuning behavior in correspondence to our simulation.

We nanofabricate PhCRs with the tantalum pentoxide ($Ta_2O_5$, hereafter tantala) material platform[33]. The PhCR coupling waveguides extend to the chip edges, enabling pump laser insertion to the chip at ~5 dB loss per facet. Figure 3b shows a section of a PhCR with a radius of 22.5 $\mu$m, obtained with a scanning electron microscope. This PhCR unit cell indicates the amplitude and period of the modulation that controls $\epsilon$ and the azimuthal mode number, respectively. Specifically, one programmed azimuthal mode is frequency shifted to higher and lower frequency resonances, separated by a photonic bandgap. By tuning the pump laser onto resonance of the lower frequency mode (hereafter the pump mode), we adjust the settings of normal GVD, $\epsilon$, and $\alpha$ to coincide with our detailed theoretical modeling. We carefully design PhCRs so that the pump mode falls within the 1550 nm wavelength range, which is convenient for commercial laser components.

Figure 3c presents our test system and procedures. We energize our devices with a tunable external-cavity diode laser amplified with an erbium-doped fiber amplifier. In our experiments, the primary observable is the soliton spectrum, which we measure with an optical-spectrum analyzer in both transmission and reflection from the chip. Assessing the soliton's noise characteristics is also important, since relatively low noise is an expected property of all the modelocked states across the dark-to-bright continuum. We photodetect the entire soliton, except the pump laser, and record the relative intensity noise with an electronic spectrum analyzer. Additionally, we use a ~150 GHz bandwidth modified uni-traveling carrier photodetector[34] and an ESA to record the outcoming pattern repetition frequency of suitable PhCR devices. See "Methods" for details.

**Exploring the continuum**. We search for dark- and bright-pulse states in normal GVD PhCRs with parameter settings derived from our theoretical model. Figure 4a indicates the detuning dependence of the transition from dark-to-bright pulses with respect to the half-filled state. In a set of experiments examining both sides of the continuum, we systematically vary $\epsilon$ with discrete PhCR devices, and we vary $\alpha$ according to the sequences $1 \rightarrow 2 \rightarrow 3$ for dark pulses or $4 \rightarrow 5 \rightarrow 6$ for bright pulses in Fig. 4a. For each setting of $\alpha$, we record the state's optical spectrum, and we directly analyze the center lobe, wing, and horn spectral signatures with respect to our theoretical predictions, identifying the dark- and bright-soliton pulse shapes in normal

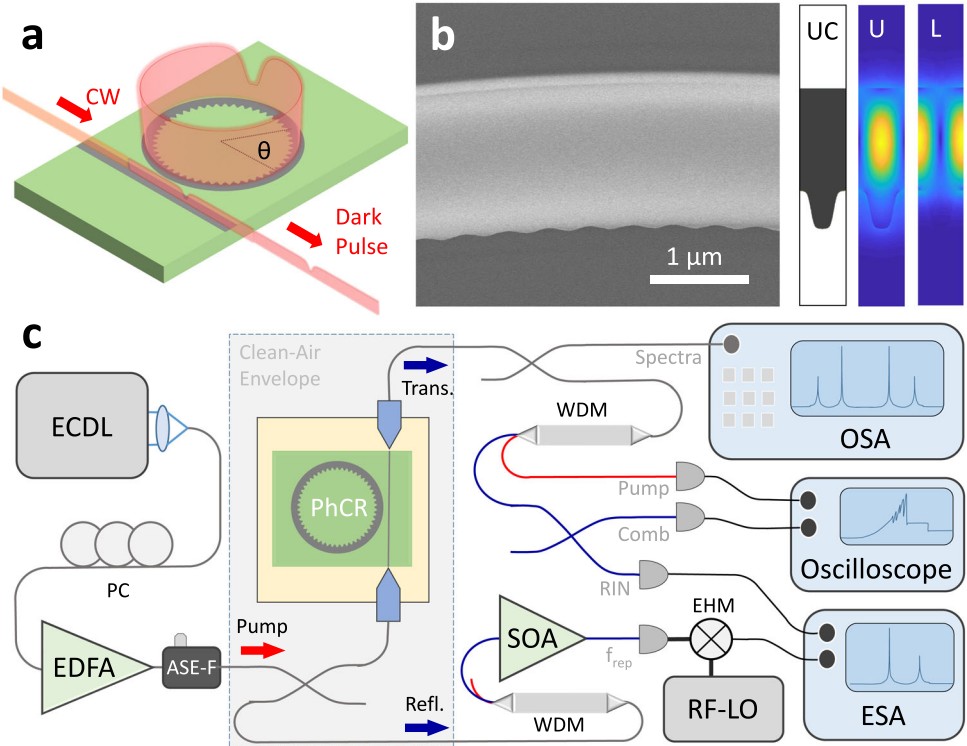

**Fig. 3 Experiment protocol. a** Illustration of dark pulse generation in the PhCR. **b** Electron microscopy image showing a section of the PhCR. The unit cell (UC) is shown on the right with exaggerated modulation for clarity. The right-most panels show simulated electric field distributions for the (U) upper and (L) lower modes. **c** Diagram showing the optical testing setup. ECDL: external-cavity diode laser, EDFA: erbium-doped fiber amplifier, ASE-F: amplified spontaneous emission filter, WDM: wavelength-division multiplexer, SOA: semiconductor optical amplifier, EHM: even harmonics mixer, LO: local oscillator, OSA/ESA: optical/electronic spectrum analyzer.

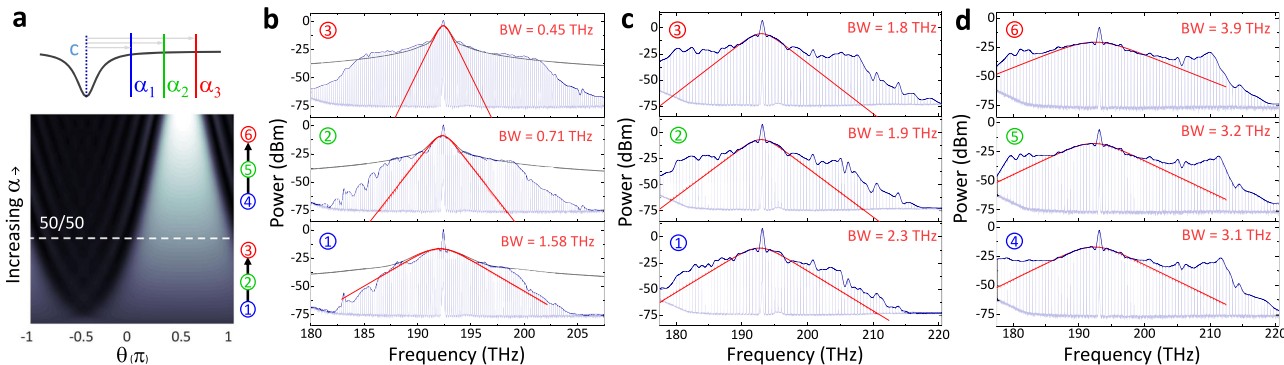

**Fig. 4 Spectral tuning. a** Illustration of a detuning sweeps toward ($1 \rightarrow 3$) and away from ($4 \rightarrow 6$) the half-filled state (white dashed line), where both sweeps are from high to low frequencies ($\alpha_1 \rightarrow \alpha_3$). The color scale is identical to Fig. 1b. **b** Measured spectra in a FSR = 200 GHz PhCR showing the center lobe bandwidth (BW, fitted red trace) decreasing with increasing detuning. The spectral envelope of a square wave (gray) is plotted for comparison. **c, d** show spectra from nominally identical FSR = 500 GHz rings, but with peak-to-peak PhC amplitude of 8 nm and 12 nm, resulting in $\epsilon = 4.5$ and $5.9$ $\kappa$, respectively. This places soliton behavior of the two PhCRs on opposite sides of the half-filled state and an opposing BW dependence on $\alpha$. Moreover, prominent horn features appear in (**d**).

GVD. In particular, we identify the reversal of spectral bandwidth tuning behavior in response to the setting of $\alpha$ on either side of the continuum; see Fig. 4a.

Figure 4b presents spectrum measurements of dark pulses. Specifically, we observe a decrease in the center-lobe bandwidth as a function of increasing $\alpha$. By fitting the center-lobe portion of the spectrum to a model proportional to $\text{sech}^2(\frac{\nu - \nu_0}{\text{BW}})$ where $\nu$ is optical frequency, $\nu_0$ is the center of the spectrum, and BW is the bandwidth, we directly characterize the center lobe. Indeed, the center-lobe bandwidth is linked to the filling fraction $t_c$ by the

expression $\text{BW} = \sqrt{3}\,\text{FSR}/\pi \cdot (1 - t_c)^{-1}$, where FSR is the free-spectral range; see Supplementary Section III. We observe a reduction in center-lobe bandwidth from 1.58 to 0.45 THz as we increase $\alpha$. In temporal units, the measurements in Fig. 4b indicate that $t_c$ varies from 0.07 to 0.25 with increasing $\alpha$, a range in agreement to our PS-LLE simulations. This data indicates a distinction of dark solitons in comparison to anomalous GVD bright solitons, which exhibit the opposite behavior with $\alpha$. Furthermore, the dark-soliton pulses develop the wing feature outside the bandwidth of the center lobe as we increase $\alpha$ and the

localized dark pulse expands into the square-wave-like pattern of the half-filled state. To highlight this behavior, we overlay in Fig. 4b the spectral envelope of a square wave with the same bandwidth of the center-lobe. The characteristic $1/\mu^2$ roll-off of the square wave reproduces the envelope of the wing feature until the bandwidth limit set by the PhCR GVD. We note that both the center lobe and wing features behave according to our theoretical prediction in Fig. 2e.

Figures 4c, d explore spectrum measurements in PhCRs designed to host dark and bright pulses, respectively. We use PhCRs with 500 GHz FSR and settings of $\epsilon = 4.5\,\kappa$ (Fig. 4c) to realize dark pulses and $\epsilon = 5.9\,\kappa$ (Fig. 4d) to realize bright pulses. In an experiment, we vary $\alpha$ according to the sequences $1 \rightarrow 2 \rightarrow 3$ for dark pulses or $4 \rightarrow 5 \rightarrow 6$ for bright pulses by monotonically tuning the pump laser frequency. By varying $\alpha$, we explore both regimes of the dark-to-bright continuum with the half-filled state as intermediate between them. Fitting the center lobe in both these regimes shows the characteristic increase in bandwidth as we vary $\alpha$ across the normal-GVD soliton continuum. Comparing Fig. 4b, c shows the continuum tuning as a general phenomenon that manifests in normal GVD resonators with significantly different repetition rates and dispersion strength. Moreover, from our theoretical predictions in Fig. 2e, we anticipate that these bright solitons (Fig. 4d) will exhibit a significant horn feature, which is the analog of more well-known dispersive waves in the anomalous-GVD regime. Our measurements demonstrate the horn feature with an ~5 dB spectral enhancement near the PhCR GVD bandwidth limit of the soliton. Similar to the dispersive waves, the horn elevates the comb power above the center-lobe envelop, leading to the observed plateau-like spectral profile characteristic of these states. The set of normal-GVD soliton spectrum measurements in Fig. 4, obtained by tuning $\alpha$, presents a comprehensive test of the dark-to-bright continuum. An additional set of spectra sweeping over the half-filled state is presented in Supplementary Section V.

**Applications**. We anticipate that nearly any state of the dark-to-bright pulse continuum will yield a useful frequency-comb source. Moreover, the normal-GVD regime of Kerr frequency combs presents unique opportunities in terms of comb lasers with designable spectral coverage, relatively constant comb-mode power distribution, and high conversion efficiency of the pump laser to the integrated comb power. Figure 5 presents examples of spectral design and noise measurements with a 200 GHz FSR PhCR.

The concept of a frequency comb is generalized from the particle-like Kerr soliton to time-stationary patterns in a resonator with a single repetition frequency. But naturally the repetition frequency and comb power vary with the parameters of the PS-LLE. Moreover, the mode-frequency splitting of our PhCRs arises from a coupling of forward and backward propagation direction, and we have observed pulse propagation both co-propagating and counter-propagating to the pump laser. The ratio of forward- and backward-propagating pump laser in the PhCR varies with $\alpha$ and $\epsilon$[35]. Each direction can form a comb, which depletes the pump power of the pump fields in both directions through the PhC coupling. This leads to competition between the forward and backward comb states. The varying pump laser ratio determines which way the pulse forms initially, while the following state competition can lead to direction-switching behaviors. The noise measurements we present here explore pulse propagation in both directions.

To characterize the repetition frequency, we operate a PhCR to generate a pulse train in opposite direction to the pump laser. For this experiment, we monitor the reverse-propagating comb power

using an optical circulator; see Fig. 5a. To measure the repetition frequency, we amplify the comb power using an optical amplifier. The comb power is delivered to a ~ 150 GHz bandwidth, 0.2 A/W responsivity photodetector[34]. The optical circulator and a 2 THz bandwidth optical filter prior to amplification and photodetection reduce photocurrent from the pump laser. We extract the 200 GHz photocurrent signal from the photodectector chip with a microwave probe, and we use a fourteenth-order harmonic mixer driven by a 14.22 GHz signal to down-convert the repetition frequency. Figure 5b shows the repetition frequency at an intermediate 175.6 MHz frequency. The high signal-to-noise ratio of the repetition frequency is consistent with a low-noise frequency comb of equidistant modes operating in the soliton regime.

In a second characterization experiment, we measure the relative intensity noise (RIN), which is a critical characteristic for example in applications that the comb modes are encoded with information. Here, we operate a PhCR to generate a pulse train in the forward direction with respect to the pump laser. The optical spectrum of a forward-emitting comb state in this measurement is shown in Fig. 5c. We separate the comb power from the transmitted pump power, using a wavelength-filtering element prior to photodetection. The photodiode has 12 GHz nominal bandwidth and 0.8 A/W responsivity to measure RIN, and we deliver the 9 mW total comb power to the detector without amplification. Figure 5d shows the (RIN) on the photodetected signal. The detector noise and the RIN of the pump laser are approximately at the same power level as the comb. The RIN level ranges from −130 dBc/Hz at 10 kHz to −160 dBc/Hz at higher frequencies, currently limited by the detector noise floor. Once the low-noise state is reached, the low RIN level is maintained over an appreciable range in $\alpha$. The measured RIN is comparable to the anomalous Kerr solitons, such as the case reported in[36].

Finally, we demonstrate how normal-GVD soliton combs in PhCRs may be used in the future. Frequency-comb lasers are revolutionizing optical communication systems, which require dense carrier grids in for example the 1300 nm and 1550 nm wavelength bands. Still, universal laser sources based on scalable photonics technology do not exist, primarily due to physical limitations of laser gain. Soliton microcombs are recognized as a promising technology for this application[7,10,11], but especially in the anomalous GVD regime there has been no demonstration of a microcomb that can support multiple wavelength bands. Here, we demonstrate a normal-GVD PhCR with suitable properties to generate a broadband comb laser with relatively constant spectral envelope and a dense 200 GHz mode spacing. Figure 5e shows the comb-laser spectrum, which spans the standardized telecommunication bands denoted U, L, C, S, E and a portion of the O band. Such a spectral coverage of 50 THz exceeds what is possible with either fiber-based solid-state gain materials or semiconductor gain materials, highlighting the uniqueness of microcomb technology. By operating in the normal GVD regime, our PhCR pulsed microcomb laser offers high conversion efficiency from the pump laser to the comb modes. Efficiency is a critical metric in for example hyperscale data centers where the demands of ever-increasing internet traffic and services causes massive energy consumption. More efficient laser sources, especially comb lasers, are one of the most important technology areas[37]. Specifically, we characterize the comb conversion efficiency $\eta = P_{comb}/F^2$ from the input pump power $F^2$, and we predict that $\eta \approx 25\%$ is attainable based on modeling with the PS-LLE. In our experiments, we obtain a conversion efficiency as high as 21% in which a PhCR converts a 33 mW pump laser to a coherent microcomb with 200 GHz mode spacing and 7 mW mode-integrated power that spans the optical frequency range from 180 THz to 210 THz.

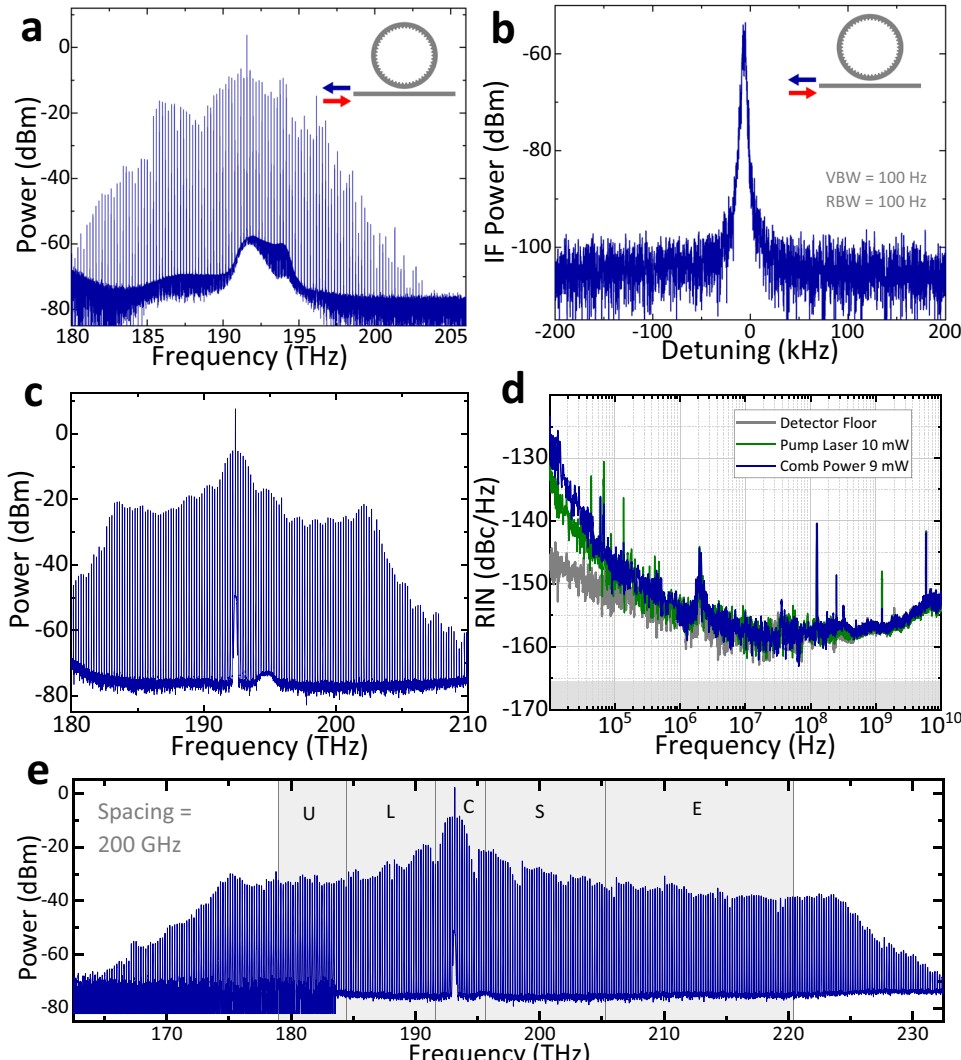

**Fig. 5 Frequency comb and noise characterization. a** Optical spectrum and (**b**) down-mixed electronic repetition rate beatnote measured at the reflection port. **c** Optical spectrum and (**d**) corresponding measured relative intensity noise on the comb power. **e** Demonstration of a 200 GHz repetition rate comb covering Telecom U through S bands.

This highlights the importance of normal GVD PhCR micro-combs for advanced optical communications technologies.

## Discussion

We have presented a regime of nonlinearity in which both bright and dark pulse states are stable in the same physical resonator. We control the balance of nonlinearity and loss to phase-match four-wave mixing in a normal-GVD PhCR by adjusting a mode bandgap for the pump laser. Moreover, these pulse states arise spontaneously from a CW-laser flat background, according to phase matching with the PhCR. Laser detuning is intrinsically linked to the intensity-filling fraction of a pulse state, and a pulse with an intensity dip can continuously evolve to a localized bright pulse. Indeed, at the center of the dark-to-bright continuum is the half-filled state, which represents the pulse transition edge between high and low-intensity levels. Both the dark and bright pulse states manifest as a frequency comb with fingerprint spectral features, which we analyze for comparison with our detailed numerical models. In particular, we expect and observe a striking inversion of pulse-bandwidth tuning with laser detuning centered on the half-filled state that highlights the fundamental

difference in the phase-matching with normal and anomalous GVD. This type of microcomb laser is a versatile and efficient multi-wavelength source with high spectral coherence, which enables various signaling and sensing applications.

## Methods

We design the resonators, including the micro-ring GVD and the photonic-crystal parameters using a commercial finite-element software. A split-step LLE program is developed in-house to simulate the nonlinear-optical states. We fabricate our device beginning with 570 nm thick ion-beam sputtered film of tantala on a 3 $\mu m$ thick oxidized silicon wafer. We anneal the tantala material in a nitrogen-oxygen gas mixture. We define the pattern for PhCRs and their waveguides using electron beam lithography, and a fluorine inductively coupled plasma reactive-ion etch (ICP-RIE) transfers the pattern to the tantala layer. We separate the wafer into chips each with several PhCR devices using a deep Si RIE. We chemically clean the chips using solvents and oxidizing agents. The chips are ready for optical testing.

We use a tunable external-cavity diode laser covering the C band to test our devices. We adjust the laser polarization by straining single-mode fiber, then amplify with an erbium-doped fiber amplifier to provide sufficient optical power for the experiments. In order to suppress amplified spontaneous emission of the amplifier, we add a tunable narrow-band filter after the amplifier. We piezoelectric tune the laser to perform the detuning sweeping in this work. We mount chips-under-test to a thermally stable platform, and we align lensed fibers to the chip for input and output. We monitor the output in transmission and reflection. We use a

fiber coupler to access the reflection port and a wavelength-division multiplexer to spectrally separate the pump laser from the transmission port. The transmitted pump laser and comb power are monitored in real-time on an oscilloscope.

We analyze the output light with an optical spectrum analyzer or by photodetection and with an electronic spectrum analyzer. The high-frequency signal from photodetecting the repetition rate is down-converted by an even-harmonic mixer driven by an RF local oscillator to a manageable frequency for the ESA. We record sets of spectra and electronic beatnotes for states at various laser detuning. See Fig. 3c for an illustration.

## Data availability

The data that support the findings of this study are available from the corresponding author upon reasonable request.

## Code availability

The simulation codes used in this study are available from the corresponding author upon reasonable request.

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

## Acknowledgements

The authors thank Jennifer Black and Travis Briles for carefully reading the paper. This research is supported by the Defense Advanced Research Projects Agency PIPES program and NIST. E.L. acknowledges support from the Swiss National Science Foundation (SNSF) under contract No. 191705.

## Author contributions

S.-P.Y. contributed the conception, design, and fabrication, and theoretical analysis; S.-P.Y., E.L., and J.Z. performed the optical and radio-frequency measurements. S.B.P. contributed to the theoretical understanding and supervised the findings of this work; All authors provided feedback and helped shape the research, analysis, and manuscript.

## Competing interests

The authors declare no competing interests.
