## [Peer review file · Nature Communications]

REVIEWERS' COMMENTS

Reviewer #1 (Remarks to the Author):

The authors thoroughly addressed the questions and requests, and they modified the paper accordingly. The revised manuscript and supplement provide sufficient details for the continuous nature of pulse states from bright to dark, which is allowed by PhCR. The content provided by this manuscript is very high quality, important to the researchers in the same field, and delivered well to attract the interests of general readers. I strongly recommend the publication of this work in Nature Communications.

Reviewer #2 (Remarks to the Author):

I have read the revised paper, supplementary information and rebuttal. I feel all technical comments have been addressed adequately and hence reproducibility of the work is ensured.

The new data included in the SI Figure 2 solves my grievance regarding the title of the manuscript. I would put it into the main manuscript, even.

The citation of existing literature has been improved and prior works are better put into context in the revised manuscript.

In summary, I fully support the publication of the revised manuscript as is in Nature Communications.

Reviewer #3 (Remarks to the Author):

I would like to thank the authors for adding additional information that clarifies the experiments and numerical calculations. However, I still believe the manuscript is not suitable for a broad audience like for Nature Communications. Another major problem is the lack of time domain measurements. Without time domain measurements it is difficult to draw conclusions from the measured spectra, in particular with the somewhat imperfect agreement between measurements and calculations.

Reviewer #1 (Remarks to the Author):

The authors thoroughly addressed the questions and requests, and they modified the paper accordingly. The revised manuscript and supplement provide sufficient details for the continuous nature of pulse states from bright to dark, which is allowed by PhCR. The content provided by this manuscript is very high quality, important to the researchers in the same field, and delivered well to attract the interests of general readers. I strongly recommend the publication of this work in Nature Communications.

We thank the reviewer for the positive remarks. The feedback in the review process undoubtedly made the manuscript more complete. We believe our work elucidates nonlinear physics and will make positive impact in the field of study and applications.

Reviewer #2 (Remarks to the Author):

I have read the revised paper, supplementary information and rebuttal. I feel all technical comments have been addressed adequately and hence reproducibility of the work is ensured.

We thank the reviewer for pointing out potential issues in our manuscript, enabling us to address them and improve the quality of our work. We believe the updated manuscript will interest readers and inspire future works.

The new data included in the SI Figure 2 solves my grievance regarding the title of the manuscript. I would put it into the main manuscript, even.

The data in Fig. S2a indeed showcases the turn-around point of the continuum tuning, in compliment to the data presented in the main text. In consideration of the length of the main text and the relatively higher complexity of the integrated spectra plots, we believe showing the current main text data while pointing interested readers to Fig. S2 would give future readers a more fluent reading experience.

The citation of existing literature has been improved and prior works are better put into context in the revised manuscript.

We thank the reviewer for gainful advice for citations, making our manuscript better.

In summary, I fully support the publication of the revised manuscript as is in Nature Communications.

We thank the reviewer for support. We hope our work to make a positive impact in publication.

Reviewer #3 (Remarks to the Author):

I would like to thank the authors for adding additional information that clarifies the experiments and numerical calculations. However, I still believe the manuscript is not suitable for a broad audience like for Nature Communications. Another major problem is the lack of time domain measurements. Without time domain measurements it is difficult to draw conclusions from the measured spectra, in particular with the somewhat imperfect agreement between measurements and calculations.

We regret to hear the reviewer's dissatisfaction with our manuscript. We respectfully disagree with the reviewer's comment favoring time-domain measurement over spectral data.

Time domain characterizations of pulsed microcombs have been well established both in the anomalous regime with the formation of dissipative soliton [T Herr et al, Nat. Photonics, 8, 145-152 (2014)], and in the normal regime with the appearance of dark pulses [X Xue et al, Nat. Photonics, 9, 594-601 (2015)]. These seminal works have shown good agreement between time- and spectral-domain measurements. With these first characterizations established, showing native FSR-spaced comb spectra with low intensity noise is generally considered good evidence for a nonlinear state.

Temporal measurements are not available in this work due to hardware limitations when our work was performed. Moreover, spectral characterization typically features a much higher dynamic range and resolution than possible with temporal characterizations and are less complex to setup and calibrate.

We believe our correspondence between simulated and measured spectra and their predicted tuning characteristics is sufficient to support the conclusions of this work. The spectra features and evolution when tuning across the described continuum agree well with the observations. We hope the reviewer would reconsider and support our work for publication.

We now discuss the temporal-spectral correspondence in our work on Page 5 Paragraph 3:

“...which we characterize through spectral-domain measurements. Time- and spectral-domain measurements show good correspondence for normal dispersion patterns [X Xue et al]. We infer time domain characteristics from the spectra and their tuning behavior in correspondence to our simulation.”

This change does address the reviewer's question in a specific fashion, putting into context how time domain measurements are typically treated in the microresonator soliton field.